# A Laser with Intracavity Spectral Beam Combining with Continuous-Wave and Pulse-Periodic Pumping of Various Lasing Channels

Vladimir Kamynin *, Vitalii Kashin, Dmitri Nikolaev, Anton Trikshev and Vladimir Tsvetkov

Prokhorov General Physics Institute of the Russian Academy of Sciences, 38 Vavilov Str., 119991 Moscow, Russia; trikshevgpi@gmail.com (A.T.); tsvetkov@lsk.gpi.ru (V.T.)
* Correspondence: kamyninva@gmail.com

**Abstract:** The output spectral, temporal, and spatial parameters of a two-channel Yb-fiber laser operating in an incoherent intracavity spectral beam-combining mode were investigated. One of the laser channels operated in continuous pumping mode, and the second channel operated in pulse-periodic pumping mode with a pulse duration of microseconds. When the channels operated separately, continuous lasing at a wavelength of $\lambda_1$ was observed in one channel, and pulsed lasing at a wavelength of $\lambda_2$ was observed in the other. When both channels operated simultaneously, it was shown that during the pump pulse action, the laser switched operation to pulsed collective mode lasing at a wavelength of $(\lambda_1 + \lambda_2)/2$. Lasing at wavelengths $\lambda_1$ and $\lambda_2$ was not observed at this time.

**Keywords:** multi-channel laser; spectral beam combining; collective lasing; fiber lasers; gain switch





## 1. Introduction

High-power laser systems are required to solve modern technological problems in the fields of precision material processing [1], advanced energetic systems [2], and the development of new radiation sources [3]. High-power laser irradiation can be achieved both by optimizing laser sources and amplifiers and by combining radiation from several sources. Modern fiber laser systems emit over several kilowatts in a single-mode regime [4]. However, further power scaling in a single-source regime can lead to strong thermal effects, low long-term stability, and inappropriate beam quality. In the case of high-beam-quality lasers, an increase in power is obtainable by laser beam combining (BC). BC methods include two large groups: coherent and incoherent methods. In both cases, an increase in lasing power and system brightness is achieved. However, in the first case, it is necessary to apply a very stable single-frequency master oscillator, and the laser system possesses a complex architecture due to the use of additional active stabilization and phase adjustment systems [5]. In addition, it is necessary to apply active feedback based on high-speed cameras or photodetectors. Over the past five years, researchers have begun to apply machine learning [6] to the management of these systems. Despite the difficulties mentioned, researchers have achieved the coherent BC of hundreds [7,8] of laser beams. Fringe contrast or combining efficiency can exceed 96%, and a phase residue error better than $\lambda/22$ has been demonstrated. Nevertheless, in all cases, some part of the energy will always be lost at the fringes. In the case of kW-level systems, this loss can be serious. The second type of source, based on incoherent BC, is less demanding of the radiation characteristics of individual laser channels, and can be created using only passive optical elements. Incoherent BC methods include the Spectral Beam Combining method (SBC). Typically, this technique involves spectrally selective elements such as diffraction gratings, dichroic mirrors, or volume Bragg gratings, and a set of independent laser sources that emit at different wavelengths. This method can be realized in several ways. The most common method is extracavity, in which the total combining of the laser beams takes place

outside of any involved laser cavity. Recent results of extracavity SBC have demonstrated output power at a level of tens of kW [9,10]. In [11], Q. Gao et al. achieved good beam quality ($M^2$ = 1.68) in a case where 32 laser channels were combined. Another approach of SBC is the intracavity combination of different wavelengths. In contrast to the extracavity approach, intracavity SBC combines different beams directly inside the laser cavity.

A number of peculiar properties of intracavity SBC have been studied in a variety of publications [12–17]. The efficiency of the system was analyzed taking into account spectral and power limitations on the number of combined laser channels. The influence of aberration on laser parameters was also investigated. However, each laser oscillator of the set was considered independently. At the same time, it is possible for some laser channels to participate in common (collective) lasing [18–23]. This type of lasing was experimentally realized and theoretically justified in [24,25]. The researchers showed that the spectral (wavelengths) and spatial (directions of beam propagation) parameters of collective lasing (CL) can be significantly different from the parameters in an SBC mode of independent channels. It was demonstrated that the occurrence of CL can lead to a parasitic effect of suppression of SBC lasing. A method of realization or suppression of the CL mode has also been proposed [25].

The goal of our experiment was to further investigate the properties of the SBC laser system. During the current investigation, we focused on the study of the conditions of CL occurrence when one of the laser channels operates in pulsed mode. To realize these conditions, both continuous-wave (CW) and pulse-periodic pumping of the active media of different channels were used simultaneously in the experiment. In this case, periodic switching of CL and SBC laser modes is possible during laser operation. This allows the spectral and spatial output parameters of the laser to be controlled over time. Laser systems with such parameters may be of interest in the fields of communications, materials processing, remote laser testing, etc.

A two-channel Yb-fiber laser was used as a model system. One lasing channel operated in continuous pumping mode. The second channel was pulse-periodic pumped. The laser cavity's schematic included a common output coupler and a diffraction grating (DG) mounted under a grazing incidence angle [12–15,17,24,25] (Littman–Metcalf scheme).

## 2. Experimental Setup

The features of the experimental optical scheme and the paths of the beams in the cavity were similar to [24,25] and are shown in Figure 1.

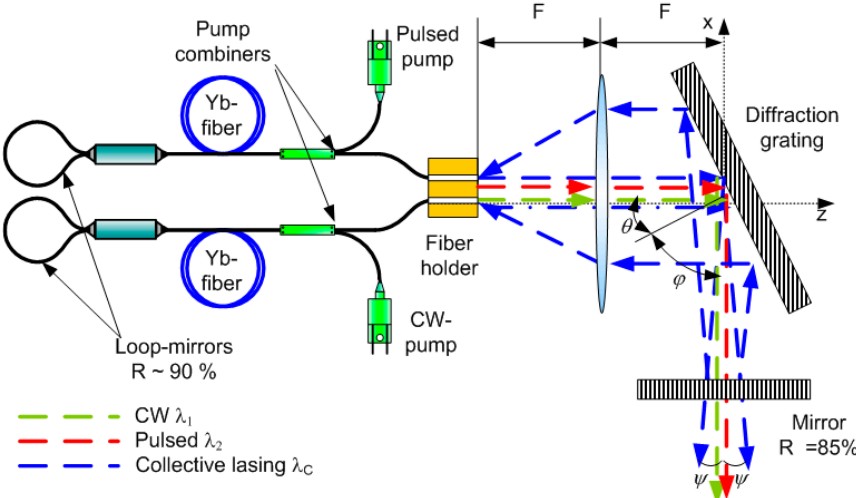

**Figure 1.** Optical scheme of the laser cavity with intracavity SBC. The upper part of the laser operates in pulsed mode. The scheme of the beams' paths is shown in different colors and line types.

Two single-mode silica glass double-clad Yb-doped fibers, 1 and 2 (lasing channels 1 and 2, respectively), were used as active media. The core and first cladding diameters of the fibers were approximately 6 and 105 μm, respectively. The numerical aperture of the fiber core was $NA_F = 0.11$. The lengths of both active fibers were the same, equaling 2 m. Active fiber pumping was performed through standard pump (2 + 1 in 1) combiners. To avoid parasitic feedback caused by Fresnel reflection, we polished the front ends of the fibers at an angle of 10°. We used a common homemade aluminum holder to fix these fibers. The axes of the fibers in the holder were parallel, and the distance between the fiber axes was set to $\Delta X \cong 125$ μm. The axes of the beams leaving both fibers were parallel to each other and to the cavity axis, located in the XY plane. The axis of beam 1 (leaving fiber 1) coincided with the axis of laser cavity Z. The axis of beam 2 was 125 microns from the cavity axis.

Fiber loop mirrors with a matching reflectivity of R = 90% were placed on the back of both active fibers. Multimode laser diodes at 976 nm wavelength were used as a pumping source. The pumping of fiber 1 was carried out in CW mode. We used a similar diode operating in a pulse-periodic mode to pump fiber 2. We achieved this mode by direct modulation of the pump current. The pump pulse duration was approximately 12 μs FWHM (Full Width at Half Maximum) and the pulse repetition rate was 1 kHz. The pump pulse shape is shown in Figure 2a. We chose fiber 2's pumping parameters in such a way that during the action of the pump pulse, the output lasing radiation of channel 2 represented a single pulse.

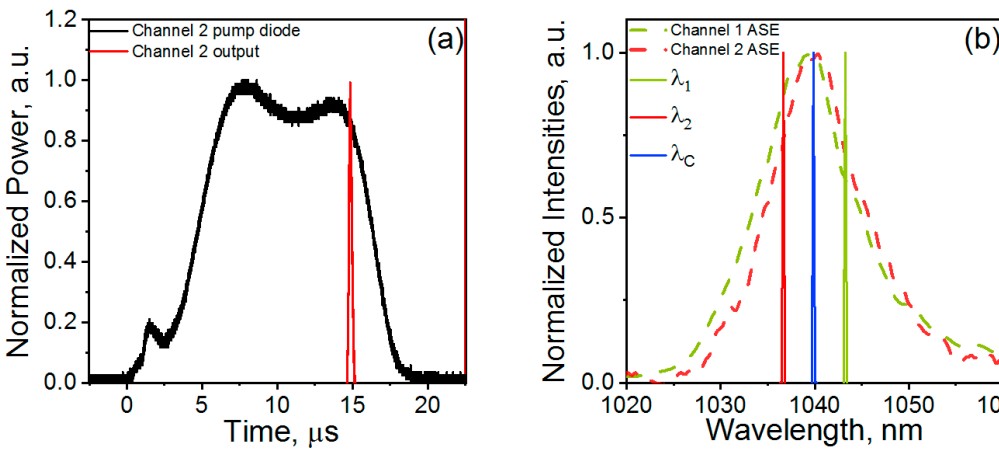

**Figure 2.** (**a**) Temporal function of channel 2 laser diode pump power (black curve), and the output pulse radiation of channel 2 (red curve). (**b**) The spectra of ASE of channel 1 and 2—green and red dashed curves, correspondingly; lasing wavelengths of channel 1 ($\lambda_1$), channel 2 ($\lambda_2$), and collective mode ($\lambda_C$)—green, red, and blue solid curves.

The spectra of amplified spontaneous emission (ASE) of active fibers 1 and 2 are shown in Figure 2b. The maximum ASE line of fiber 1 was 1039 nm, and the linewidth was about 11 nm FWHM. The maximum ASE line of fiber 2 was 1040 nm and also 11 nm FWHM. Because the ASE spectra of both active fibers were almost identical, the choice of CW and pulse operation modes of channels 1 and 2 was arbitrary. Inverting channel-operating modes should not affect the parameters of collective lasing.

The front end faces of the fibers were settled down in a focal plane of positive lens F with a focal length equal to F = 55 mm. This lens transformed initial beams into Gaussian beams with total divergence equal to $2V \approx 5 \times 10^{-5}$ rad and waist $\omega \approx 6.3$ mm at the $1/e^2$ level. Further, these beams were directed to the inclined diffraction grating with a groove density of N = 300 g/mm. Diffracted beams fell on a plane output mirror ($R_{out} = 85\%$ in a spectral range of 1050–1100 nm) at a normal angle. The distance from the DG to the output mirror was 85 mm. The point of intersection of the DG surface with the optical axis of cavity Z and the optical axes of the central rays was located in the right focal plane of lens F.

The sum of the incidence angles θ an angle between an axis Z and the normal to DG) and of the diffraction angle φ (diffraction order equal to −1) θ + φ = 90° at a laser wavelength equal to 1043.2 nm. The values of θ = 32.21° and φ = 57.79° corresponded to these requirements.

The axis of beam 1 coincided with the laser cavity axis Z (see Figure 1), and the transverse coordinate of fiber 1 was $X_1 = 0$. Therefore, the wavelength $\lambda_1$ was equal to 1043.2 nm. The wavelength $\lambda_2$ depends on the transverse coordinate $X_2$ of fiber 2 and is equal to [24]:

$$\lambda_2 = d(\sin(\theta + \text{arctg}(X_2/F)) - \sin\varphi)/m_1, \tag{1}$$

where d = 1/N, F = 55 mm (focal length of positive lens F), and $m_1 = -1$ (diffraction order). Taking into account these data, the calculated value of wavelength $\lambda_2$ was obtained as 1036.8 nm. Both wavelengths $\lambda_1$ and $\lambda_2$ lay symmetrically with respect to the maxima of the ASE spectra of active fibers 1 and 2 (Figure 2b).

The wavelength of the collective beam $\lambda_C$ should fall between wavelengths $\lambda_1$ and $\lambda_2$ and should be equal to [24]:

$$\lambda_c \cong (\lambda_1 + \lambda_2)/2 \cong 1040 \text{ nm.} \tag{2}$$

This value roughly corresponds to the maxima of the gain lines of the active fibers (see Figure 2b ASE spectra).

The collective laser emission consists of two beams (Figure 1) that propagate at angles ±ψ with respect to the axis of the laser cavity. These angles are equal to [24,25]:

$$\psi(\lambda_c) = \varphi \pm \varphi_c(\lambda_c), \tag{3}$$

where

$$\varphi_c(\lambda_c) = \arcsin(\sin(\theta_o + \text{arctg}(X_1/F)) - m_1\lambda_c/d). \tag{4}$$

Under our experimental conditions and with $X_1 = 0$, $\lambda_1 = 1043.2$ nm, and $m_1 = -1$, the tilt angles of the output collective lasing beams were ψ ≅ ±1.8 mrad.

The total diffraction efficiency of the DG used was about 60%. As a result, the percentage of SBC radiation was only 15% of the total output power of the laser. Another part of the radiation was extracted through other diffraction orders of the DG. As the diffraction efficiency of the DG increases, the proportion of SBC radiation will increase.

The spectra of the laser output were studied using a spectrum analyzer (Avesta ASP-150C). The spectral range of the analyzer was 460–1100 nm, and the resolution was equal to 0.4 nm. The minimum signal exposure time was 7 ms. This time corresponded to seven periods of pump pulses.

The zero order of the DG was used to investigate the output emission spectra. The lasing temporal parameters were studied using a 4-channel oscilloscope with 4 GHz analog bandwidth and fast photodiodes with a resolution of 0.2 ns. The spatial separation of output beams 1 and 2 and the collective beam was carried out using two crossed diffraction gratings (two sequentially installed diffraction gratings with perpendicularly oriented diffraction grating lines). As a result, we can simultaneously observe independent dynamics in different spectral channels corresponding to channel 1 ($\lambda_1$), channel 2 ($\lambda_2$), and $((\lambda_1 + \lambda_2)/2)$.

## 3. Results

First, the characteristics of each channel were studied separately. In this case, in the first channel, there was CW lasing at wavelength $\lambda_1$ (Figure 3a) (while the second channel was not active), and in the second channel, there was pulsed lasing at wavelength $\lambda_2$ (Figure 3b) (while the first channel was off).

During the action of the pump pulse, the output lasing radiation of channel 2 represented a single pulse. The oscilloscope trace of channel 2 lasing (fiber 1 was not pumped) is shown in Figure 2a. The pulse duration was approximately 0.2 μs.

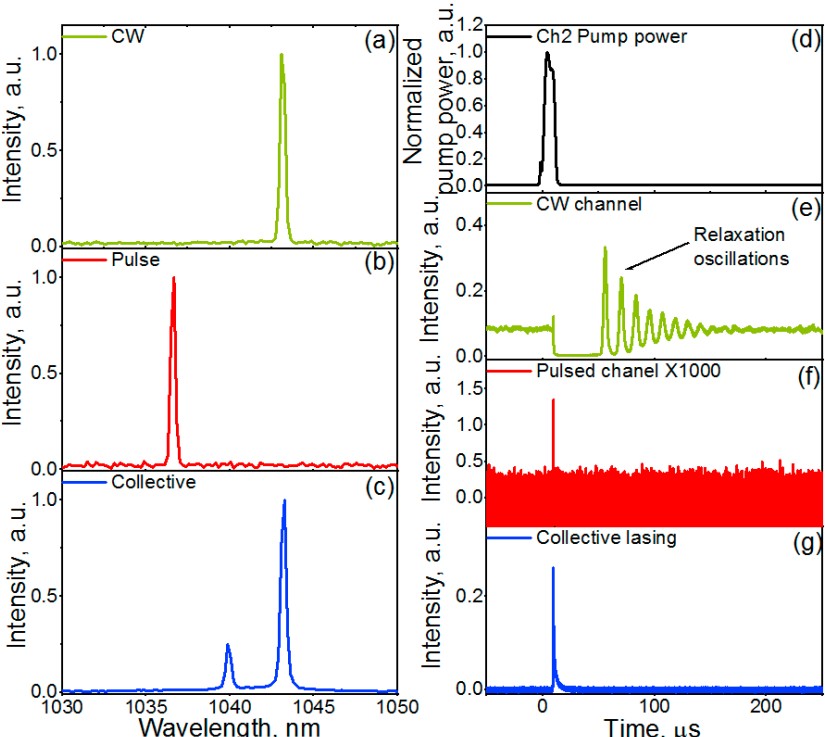

**Figure 3.** (**a**–**c**) Spectra of the output laser radiation: (**a**) channel 1 is active only; (**b**) channel 2 is active only; (**c**) both channels are active. (**d**–**g**) Oscilloscope traces of laser output emission when both channels are active: (**d**) ch2 laser diode pump power; (**e**–**g**) radiation dynamics at $\lambda_1$, $\lambda_2$, and $\lambda_c$, correspondingly. For $\lambda_2$, we show an enlarged oscilloscope waveform to clarify the residual pulse from channel 2 above the detector noise.

Oscilloscope traces of the joint operation of all laser channels are shown in Figure 3e–g. As follows from the presented results, collective lasing took place during the pumping pulse of fiber 2. At other times, lasing of channel 1 at a wavelength of $\lambda_1$ was observed. Lasing of channel 2 at a wavelength of $\lambda_2$ was not observed during laser output (see Figure 3c). The duration of the collective pulse was approximately 0.5 µs (see Figure 3g). This pulse was twice as long as the pulse from channel 2, and it has an asymmetrical shape.

The spectra of the laser output emissions are shown in Figure 3a–c. The maximum of the separate CW-lasing spectral line of channel 1 (Figure 3a) was at $\lambda_1$ = 1043.2 nm. The maximum of the separate pulse-lasing spectral line of channel 2 (Figure 3b) was at $\lambda_2$ = 1036.8 nm. The measured values of the laser linewidth were about 0.4 nm. However, this value is consistent with the instrument function of the spectrum analyzer used. The calculated value of linewidth was identical for all lines and was about 0.2 nm. In the case of both channels, simultaneous operation of two spectral lines was observed in the laser output emission spectrum (Figure 3c) with maxima at 1043.2 nm and at 1040 nm. The first line corresponds to the wavelength $\lambda_1$ of the independent lasing of channel 1, and the second line corresponds to the wavelength $\lambda_C$ of the collective lasing. No spectral line at wavelength $\lambda_2$ was observed. The observed spectrum corresponded to a long exposure time of our spectrum analyzer.

The oscilloscope traces of the laser channel operation are shown in Figure 3d–g. The results of the experiment show that a collective lasing pulse is generated during the action of the pump pulse of channel 2. At the same time, the continuous lasing of channel 1 is suppressed for 50 µs. It can be seen from Figure 3e that the suppression level of CW is 10 times or more. After this time, relaxation oscillations take place, and then the emission is converted back to CW laser light 200 µs after the pulse from channel 2. The relaxation oscillation pulse period was 15 µs. The duration of the CW lasing suppression time in our case is related to the accumulation time of population inversion in active fiber 1 up to its

threshold value. Thus, during the pulse pumping of channel 2, the total pumping power of both channels is directed at collective radiation lasing only. The results of calculations demonstrated that such behavior of the laser is explained by the fact that the collective mode lasing threshold (due to the gain coefficients of active media 1 and 2 acting together) is almost two times lower than the thresholds of independent lasing of channels 1 and 2 [25]. Under our experimental conditions, the threshold gain of fiber 2 during channel 2 independent lasing (as well as fiber 1 in channel 1) is G = 1.91 per pass through the active fiber. Under collective lasing, the threshold gain of active fiber 2 is about G = 1.08. In the last case, it is assumed that the threshold of independent lasing in channel 1 has been achieved.

If we compare the output pulse directly from channel 2 (while channel 1 is unpumped) with the pulse we obtained at the collective lasing wavelength, we can see that the pulse shapes and durations are significantly different. To understand the physical background of these processes, we need to perform numerical simulations. And this will be the goal of our future research.

When both channels were active, the laser output radiation was realized by two beams propagating at angles of $\pm(1.8 \pm 0.2)$ mrad with respect to the axis of the laser cavity. These values are in good agreement with the values predicted by Equation (3) for the case of collective mode lasing.

We believe that it is possible to achieve a pure pulse-periodic regime using this technique in cases where channel 2 is modulated at higher frequencies (20 kHz or higher). Moreover, in this situation, channel 1 can be used as an additional gain medium for collective lasing. This allows higher pulse energies to be achieved in gain switch mode. In addition, we consider this research promising for the study of collective lasing instabilities.

## 4. Conclusions

The characteristics of the operation of a two-channel laser with intracavity spectral combining of laser channels with CW and pulse-periodic pumping of different laser channels were investigated. We showed that during the pump pulse action in one of the laser channels, the laser was switched from CW mode operation to pulse laser mode with collective mode pulse output. This behavior of the laser is explained by the fact that the collective mode lasing threshold (due to the gain coefficients of active fibers 1 and 2 acting together) is almost two times lower than the threshold gain coefficients of channels 1 and 2 during independent lasing. When using a number of laser channels with CW pumping, it is possible to periodically switch the laser operating modes between the SBC mode and the lasing of a collective mode pulse. The output parameters of collective lasing are essentially different from the parameters in an SBC mode of independent channels. Thus, by using pulse-periodic pumping in one of the lasing channels, periodic modulation of the output spectral (lasing wavelengths) and spatial (directions of beam propagation) parameters of the entire laser is possible.

**Author Contributions:** Conceptualization, V.K. (Vladimir Kamynin), D.N. and V.T.; methodology, D.N.; investigation, D.N. and V.K. (Vladimir Kamynin); data analysis, D.N.; writing—original draft preparation and editing D.N., V.K. (Vladimir Kamynin) and V.T.; visualization, D.N.; resources, D.N., V.K., A.T. and V.K. (Vitalii Kashin); and project administration, V.T. All authors have read and agreed to the published version of the manuscript.

**Funding:** This research was funded with the financial support of the Ministry of Science and Higher Education of the Russian Federation, grant number No. 075-15-2022-315, and carried out on the basis of the World-Class Research Center «Photonics».

**Institutional Review Board Statement:** Not applicable.

**Informed Consent Statement:** Not applicable.

**Data Availability Statement:** The data supporting the findings of this study are available within the article.

**Conflicts of Interest:** The authors declare no conflict of interest.

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
