# Peer review of "A Laser with Intracavity Spectral Beam Combining with Continuous-Wave and Pulse-Periodic Pumping of Various Lasing Channels"

_photonics, doi:10.3390/photonics10121389_

Round 1
Reviewer 1 Report
Comments and Suggestions for Authors
Author Response
Dear Reviewer!
We are deeply grateful to you for the consideration of possibility of publishing of the article in your journal and for your work to improve the quality of the article. We give thanks the referees for the comments.
We take into account all the remarks of referees.
Please find the list of remarks (bold) and our answers and corrections (Italic) below.
Sincerely yours
Vladimir A. Kamynin
(1) In the introduction, the experimental phenomena and data obtained by other teams using the Spectral Beam Combining method should be reviewed in more detail.
We have expand the introduction and add some reference.
(2) Spectral Beam Combining method will increase the complexity of the laser system, so will the total power of the laser be affected due to beam combining and the loss of optical components ?
Thank you for your comment. We have added text to the article body (page 4):
The total diffraction efficiency of used DG was about 60%. “As a result, the percentage of SBC radiation was only 15% of the total output power of the laser. Another part of the radiation was extracted through other diffraction orders of DG. As the diffraction efficiency of the DG increases, the proportion of SBC radiation will increase.”
(3) Does the temperature of the laser affect the SBC system ? Is it necessary to take effective temperature stabilization measures to ensure the stability of the SBC system ?
The affect of laser temperature on the parameters of incoherent SBC is weak. In our case, temperature stabilization is not required. This is an advantage of this method. We noted this in the introduction….” The second type of sources based on incoherent BC is less demanding of the radiation characteristics for individual laser channels and can be created while using only passive optical elements. Incoherent BC methods include the Spectral Beam Combining method (SBC).”
(4) The format of the formulas in the manuscript should be consistent.
We have corrected it.

Reviewer 2 Report
Comments and Suggestions for Authors
In the manuscript ' Laser with Intracavity Spectral Beam Combining with CW and Pulse-periodic Pumping of Various Lasing Channels,' the authors Kamynin et al reported their research applying a compact-structure Yb-fiber laser system outputting multi-channel lasers with the wavelengths of 1040nm, 1043nm and 1036nm which is a very interesting and useful work in some promising fields such as the medical imaging. I think this manuscript can be accept as the publication in photonics after the following questions are answered.
1. The authors should give the power information of each channel.
2. The ratio between laser and ASE was not given which will generate the confusion in the fig2(b). The authors should review this part.
3. The authors should give some applications of this laser system.
Author Response
Dear Reviewer!
We are deeply grateful to you for the consideration of possibility of publishing of the article in your journal and for your work to improve the quality of the article. We give thanks the referees for the comments.
We take into account all the remarks of referees.
Please find the list of remarks (bold) and our answers and corrections (Italic) below.
Sincerely yours
Vladimir A. Kamynin
- The authors should give the power information of each channel. The main goal of our researcher was to show the lasing switching effect. So all information was given in normalized units. Under present conditions we observed power levels about of 10 mW.
- The ratio between laser and ASE was not given which will generate the confusion in the fig2 (b). The authors should review this part.
Axes labels were corrected to normalized intensities (see new fig.2).
- The authors should give some applications of this laser system.
Thank you for your comment. We have added text to the article body (introduction):
“Laser systems with such parameters may be of interest in the fields of communications, material processing, remote laser testing, etc.”

Reviewer 3 Report
Comments and Suggestions for Authors
Please see comments in attached file

Comments on the Quality of English Languagegrammatical corrections are included in the attached file
Author Response
Dear Reviewer!
We are deeply grateful to you for the consideration of possibility of publishing of the article in your journal and for your work to improve the quality of the article. We give thanks the referees for the comments.
We take into account all the remarks of referees.
Please find the list of remarks (bold) and our answers and corrections (Italic) below.
Sincerely yours
Vladimir A. Kamynin
REMARKS:
Page 1.
- only' bold claim! Perhaps delete this work to not be proven wrong. For example, one could argue that a chain of amplifiers could be used.
Thank you for your remark. We removed the word “only”.
- beam
We have corrected it.
- source is
We have corrected it.
- 4. You use intracavity and extracavity in these two sentences. Is intracavity an error or did you intend to point out the two as different? If intracavity is intended, it should have its own reference if you want to mention both.
We corrected the text.
Page 2.
- Please point out what is new her compared to your previous work in refs 19-20.
We have added text to the article body “This may allow us to control the spectral and spatial output parameters of the laser over time.”
- I prefer the diagram of the fiber holder and the beam paths in Fig 1 of ref 20 better. It shows the different paths each laser beam takes in the resonator better.
Thanks for the comment. You're probably right, but we wouldn't want to repeat the same figure in different articles.
- A
We have corrected it.
- a
We have corrected it.
Page 3.
- m_1
m1
- lay
We have corrected it.
3.be
Has been deleted.
Page 4.
- inclination angles of the output collective beams are y @ ±1.8 mrad.
It was corrected.
2.with
Has been deleted.
- with the
a
- I think this should be Figure 3(b).
Figure 2(a)
Page 5.
- This paragraph is somewhat redundant with the previous section 3 results.. Pleases revise and condense.
This paragraph was condensed.
- It would be helpful to include a table of calculated and experimentally measured values of all parameters that were compared.
- the
Has been deleted
- What fraction of the total pump power goes into the collective mode? How does that compare to the efficiencies of the two channels operating alone?
Page 6: …”during the pulse pumping of channel 2, the total pumping power of both channels is directed at collective radiation lasing only.”
The study of efficiency was not the purpose of this work. However, it was shown in [19] that under close conditions the efficiency of collective generation is 1.5 - 2 times higher than that of independent generation of laser channels.
- angles of ± (1.8±0.2) mrad (suggest you refer to the calculated angle and the good agreement you have)
When both channels were active, the laser output radiation was realized by two beams propagating at angles of ± (1.8±0.2) mrad with respect to the axis of the laser cavity. These values are in good agreement with the values predicted by equation (3) for the case of collective mode lasing.
- Suchlike
Such like
- periodically
It was corrected.

Reviewer 4 Report
Comments and Suggestions for Authors
The manuscript “Laser with Intracavity Spectral Beam Combining with CW and Pulse-periodic Pumping of Various Lasing Channels” by V. Kamynin et al. deals with the generation of the central wavelength of two laser cavities with intracavity SBC by using pulsed pumping for one of the fibers. This topic is interesting for publication in Photonics.
However, several aspects of the manuscript need improvements.
1. One of the key features is the emission at the collective wave, while the individual wavelength l1 and l2 are suppressed. But in case of the cw pumped laser only for a short time. The authors should guide to possible real world applications for their research and this experiment in the introduction.
Despite the photo diode measurements showing the suppression for the cw pumped l1, spectral measurements show both lc and l1. The authors should discuss a plan to investigate the dynamics with higher temporal resolution in spectral domain (perhabs with a Pockels cell or an AOM in front of the spectrometer) as an outlook Also a spectrometer with higher spectral resolution in the desired wavelength range should be used. Maybe a Czerny-Turner spectrograph with a larger groove density and a fast camera as detector (30 µs) can give further insides.
2. Is there a dependence on the pump current and pump pulse duration of the periodic pulsed pump of channel 2? Can the output pulse duration of channel 2 be modified? How would this influence the suppression of l1? What happens if the situation is reversed (Channel 1 pulsed pump, channel 2 cw)?
3. Figure 2b: Here clearly the resolution limit of the spectrometer can be seen. The measurements should be done with better spectral resolution.
4. Formula 1-4: Some are bold. Also their indention is different. This should be unified.
5. Line 36 and others: “Various aspects of SBC at the implementation of a laser set being placed in the common cavity were considered in different publications [7-12].” The authors should explain the development of the research field in more detail according to the author instructions. “The introduction should briefly place the study in a broad context and highlight why it is important. It should define the purpose of the work and its significance, including specific hypotheses being tested. The current state of the research field should be reviewed carefully and key publications cited. Please highlight controversial and diverging hypotheses when necessary. Finally, briefly mention the main aim of the work and highlight the main conclusions. Keep the introduction comprehensible to scientists working outside the topic of the paper. “
6. Line 64: NA_F=0.11
Author Response
Dear Reviewer!
We are deeply grateful to you for the consideration of possibility of publishing of the article in your journal and for your work to improve the quality of the article. We give thanks the referees for the comments.
We take into account all the remarks of referees.
Please find the list of remarks (bold) and our answers and corrections (Italic) below.
Sincerely yours
Vladimir A. Kamynin
The manuscript “Laser with Intracavity Spectral Beam Combining with CW and Pulse-periodic Pumping of Various Lasing Channels” by V. Kamynin et al. deals with the generation of the central wavelength of two laser cavities with intracavity SBC by using pulsed pumping for one of the fibers. This topic is interesting for publication in Photonics.
However, several aspects of the manuscript need improvements.
Thank you for your remarks. During the work we corrected text and have added text to the article body
- One of the key features is the emission at the collective wave, while the individual wavelength l1 and l2 are suppressed. But in case of the cw pumped laser only for a short time. The authors should guide to possible real world applications for their research and this experiment in the introduction.
Introduction: …” In this (our) case, the periodic switching of CL and SBC lasing modes is possible at the laser operation. This may allow us to control the spectral and spatial output parameters of the laser over time. Laser systems with such parameters may be of interest in the fields of communications, medical imaging, remote laser testing, etc.”
Despite the photo diode measurements showing the suppression for the cw pumped l1, spectral measurements show both lc and l1.
In body of the article: …”We used a similar diode operating in a pulse-periodic mode to pump fiber 2. The pulse duration was approximately 12 μs FWHM and the pulse repetition rate was 1 kHz.”…” The spectra of the laser output were studied using a spectrum analyzer Avesta ASP-150C. The spectral range of the analyzer was 460-1100 nm, and the resolution was equal to 0.4 nm.The minimum signal exposure time was 7 ms. This time corresponded to seven periods of pump pulses.”
Therefore, during spectral measurements, we observed both the spectral line of CW lasing of channel 1 and the spectral line of collective lasing. We couldn't separate it in time.
The authors should discuss a plan to investigate the dynamics with higher temporal resolution in spectral domain (perhabs with a Pockels cell or an AOM in front of the spectrometer) as an outlook Also a spectrometer with higher spectral resolution in the desired wavelength range should be used. Maybe a Czerny-Turner spectrograph with a larger groove density and a fast camera as detector (30 µs) can give further insides.
Thank you very much for the discussion. This is a theme for further research.
- Is there a dependence on the pump current and pump pulse duration of the periodic pulsed pump of channel 2? Can the output pulse duration of channel 2 be modified? How would this influence the suppression of l1? What happens if the situation is reversed (Channel 1 pulsed pump, channel 2 cw)?
We have added text(page 4, Paragraphs 1,2): “We chose the fiber 2 pumping parameters in such a way that during the action of the pump pulse, the output lasing radiation of channel 2 rep-resented a single pulse.”… “ Due to the fact that the ASE spectra of both active fibers were almost identical, the choice of CW and pulse operation modes of channels 1 and 2 was arbitrary. Inverting channel operating modes should not affect the parameters of collective lasing.”
(Page 5): “The duration of the CW lasing suppression time in our case is related to the accumulation time of the inverse population in active fiber 1 till its threshold value.”
- Figure 2b: Here clearly the resolution limit of the spectrometer can be seen. The measurements should be done with better spectral resolution.
Thank you very much for the discussion. This is a topic for further work.
- Formula 1-4: Some are bold. Also their indention is different. This should be unified.
It was corrected.
- Line 36 and others: “Various aspects of SBC at the implementation of a laser set being placed in the common cavity were considered in different publications [7-12].” The authors should explain the development of the research field in more detail according to the author instructions. “The introduction should briefly place the study in a broad context and highlight why it is important. It should define the purpose of the work and its significance, including specific hypotheses being tested. The current state of the research field should be reviewed carefully and key publications cited. Please highlight controversial and diverging hypotheses when necessary. Finally, briefly mention the main aim of the work and highlight the main conclusions. Keep the introduction comprehensible to scientists working outside the topic of the paper. “
We added new references and discussion to the introduction.
- Line 64: NA_F=0.11
It was corrected.

Reviewer 5 Report
Comments and Suggestions for Authors
The manuscript (photonics-2733916 by V. Kamynin et al.) describes spectral beam combining results utilizing Yb-fibers for continuous-wave and pulsed pumping. The manuscript shows various lasing channels and their dynamics when they operate separately and simultaneously. The spectrum and oscillogram are analyzed in detail. The results will be interesting to readers in the related community, and the manuscript can be improved before publication in Photonics including the following comments.
1) For the introduction part, as laser sources by combining several sources are mentioned and are very critical to describe general and recent technologies in line ~23-26, it is recommended to include the promising device to combine light sources compactly [Laser Photonics Rev 16(4) 2100501 (2022).]. The recently reported waveguide beam splitting/combining component must be interesting for the readers so that can make this manuscript would have impact and compelling.
2) In the manuscript, FWHM is not abbreviated, so it could be added in the revised manuscript. And It is recommended to re-read the manuscript thoroughly to correct any errors/typos.
3) For colors in the figures, channel 2 and collective one are not so distinguishable. To clearly indicate the figures, it is recommended to change colors in graphs/figures more visibly for better understanding and readability for readers and impact of the results of this manuscript.
Author Response
We are deeply grateful to you for the consideration of possibility of publishing of the article in your journal and for your work to improve the quality of the article. We give thanks the referees for the comments.
We take into account all the remarks of referees.
Please find the list of remarks (bold) and our answers and corrections (Italic) below.
Sincerely yours
Vladimir A. Kamynin
1) For the introduction part, as laser sources by combining several sources are mentioned and are very critical to describe general and recent technologies in line ~23-26, it is recommended to include the promising device to combine light sources compactly [Laser Photonics Rev 16(4) 2100501 (2022).]. The recently reported waveguide beam splitting/combining component must be interesting for the readers so that can make this manuscript would have impact and compelling.
We added some new promising references to the introduction.
2) In the manuscript, FWHM is not abbreviated, so it could be added in the revised manuscript. And It is recommended to re-read the manuscript thoroughly to correct any errors/typos.
We fixed it. The first mention of FWHM: “The pulse duration was approximately 12 μs FWHM (Full Width at Half Magnitude) and the pulse repetition rate was 1 kHz.”
3) For colors in the figures, channel 2 and collective one are not so distinguishable. To clearly indicate the figures, it is recommended to change colors in graphs/figures more visibly for better understanding and readability for readers and impact of the results of this manuscript
We've updated all the drawings

Round 2
Reviewer 4 Report
Comments and Suggestions for Authors
The authors have improved the manuscript according to the suggestions of reviewes. They satisfactorily answered all remarks of the reviewer reports.
Author Response
Thank you for your comment.